# Chemistry and the Potential Antiviral, Anticancer, and Anti-Inflammatory Activities of Cardiotonic Steroids Derived from Toads [note 1]

**DOI:** 10.3390/molecules27196586

**Published:** 2022-10-05

**Authors:** Hesham R. El-Seedi, Nermeen Yosri, Bishoy El-Aarag, Shaymaa H. Mahmoud, Ahmed Zayed, Ming Du, Aamer Saeed, Syed G. Musharraf, Islam M. El-Garawani, Mohamed R. Habib, Haroon Elrasheid Tahir, Momtaz M. Hegab, Xiaobo Zou, Zhiming Guo, Thomas Efferth, Shaden A. M. Khalifa

**Affiliations:** 1International Research Center for Food Nutrition and Safety, Jiangsu University, Zhenjiang 212013, China; 2Pharmacognosy Group, Department of Pharmaceutical Biosciences, Biomedical Centre, Uppsala University, Box 591, SE 751 24 Uppsala, Sweden; 3International Joint Research Laboratory of Intelligent Agriculture and Agri-Products Processing (Jiangsu Education Department), Zhenjiang 212013, China; 4Department of Chemistry, Faculty of Science, Menoufia University, Shebin El-Kom 32512, Egypt; 5Chemistry Department of Medicinal and Aromatic Plants, Research Institute of Medicinal and Aromatic Plants (RIMAP), Beni-Suef University, Beni-Suef 62514, Egypt; 6School of Food and Biological Engineering, Jiangsu University, Zhenjiang 212013, China; 7Biochemistry Division, Chemistry Department, Faculty of Science, Menoufia University, Shebin El-Koom 32512, Egypt; 8Division of Chemistry and Biotechnology, Graduate School of Natural Science and Technology, Okayama University, Okayama 7008530, Japan; 9Center for Targeted Drug Delivery, Department of Biomedical and Pharmaceutical Sciences, Chapman University School of Pharmacy, Harry and Diane Rinker Health Science Campus, Irvine, CA 92618, USA; 10Zoology Department, Faculty of Science, Menoufia University, Shebin El-Kom 32512, Egypt; 11Pharmacognosy Department, College of Pharmacy, Tanta University, Elguish Street, Tanta 31527, Egypt; 12National Engineering Research Center of Seafood, School of Food Science and Technology, Dalian Polytechnic University, Dalian 116024, China; 13Department of Chemistry, Quaid-i-Azam University, Islamabad 45320, Pakistan; 14H.E.J. Research Institute of Chemistry, International Center for Chemical and Biological Sciences (ICCBS), University of Karachi, Karachi 75270, Pakistan; 15Medical Malacology Department, Theodor Bilharz Research Institute, Imbaba, Giza 12411, Egypt; 16Department of Pharmaceutical Biology, Institute of Pharmaceutical and Biomedical Sciences, Johannes Gutenberg University, 55128 Mainz, Germany; 17Department of Molecular Biosciences, The Wenner-Gren Institute, Stockholm University, SE 10691 Stockholm, Sweden

**Keywords:** anticancer, antiviral, anti-inflammatory, cardiotonic steroids, bufadienolides, bufotalin, bufalin

## Abstract

Cardiotonic steroids (CTS) were first documented by ancient Egyptians more than 3000 years ago. Cardiotonic steroids are a group of steroid hormones that circulate in the blood of amphibians and toads and can also be extracted from natural products such as plants, herbs, and marines. It is well known that cardiotonic steroids reveal effects against congestive heart failure and atrial fibrillation; therefore, the term "cardiotonic" has been coined. Cardiotonic steroids are divided into two distinct groups: cardenolides (plant-derived) and bufadienolides (mainly of animal origin). Cardenolides have an unsaturated five-membered lactone ring attached to the steroid nucleus at position 17; bufadienolides have a doubly unsaturated six-membered lactone ring. Cancer is a leading cause of mortality in humans all over the world. In 2040, the global cancer load is expected to be 28.4 million cases, which would be a 47% increase from 2020. Moreover, viruses and inflammations also have a very nebative impact on human health and lead to mortality. In the current review, we focus on the chemistry, antiviral and anti-cancer activities of cardiotonic steroids from the naturally derived (toads) venom to combat these chronic devastating health problems. The databases of different research engines (Google Scholar, PubMed, Science Direct, and Sci-Finder) were screened using different combinations of the following terms: “cardiotonic steroids”, “anti-inflammatory”, “antiviral”, “anticancer”, “toad venom”, “bufadienolides”, and “poison chemical composition”. Various cardiotonic steroids were isolated from diverse toad species and exhibited superior anti-inflammatory, anticancer, and antiviral activities in in vivo and in vitro models such as marinobufagenin, gammabufotalin, resibufogenin, and bufalin. These steroids are especially difficult to identify. However, several compounds and their bioactivities were identified by using different molecular and biotechnological techniques. Biotechnology is a new tool to fully or partially generate upscaled quantities of natural products, which are otherwise only available at trace amounts in organisms.

## 1. Introduction

Cardiotonic steroids (CTS) were documented by ancient Egyptians more than 3000 years ago [1]. It is well known that cardiotonic steroids show therapeutic activity against congestive heart failure; therefore, the term "cardiotonic" has been coined. Cardenolides can also be used to treat atrial fibrillation by increasing intracellular Na^+^ and controlling the contraction of cardiac fibers [2]. The sodium pump is composed of two subunits in equimolar ratios: (i) the α-catalytic subunit which is a multipass transmembrane protein containing the binding sites for Na^+^, K^+^, ATP, and CTS, and (ii) the β regulatory subunit, a transmembrane protein with several glycosylation sites [3]. The positive inotropic activity of cardiotonic steroids that mediate clinically useful physiological effects in patients has been attributed largely to high-affinity inhibitory interaction with the extracellular surface of the membrane-bound sodium pump (Na^+^/K^+^-ATPase) [4]. However, CTS were first reported to be vital in the regulation of renal sodium transport and arterial pressure. Recent epidemiological evaluations have implicated that these types of steroids have an intrinsic role in the regulation of cell growth, differentiation, apoptosis, and fibrosis, the modulation of immunity, and the control of various central nervous functions and even behavior [5]. According to Dvela et al., compounds that are derivatives from CTS have been used for centuries to treat cardiac failure and arrhythmias in Western and Eastern medicine [6]. CTS also contribute to pro-hypertrophic and pro-fibrotic cell signaling [7]. CTS are divided into two distinct groups, i.e., cardenolides, such as ouabain and digoxin, and bufadienolides, such as marinobufagenin, telocinobufagin, and bufalin [8], as shown in Figure 1. In general, herbs and animal preparations containing CTS have been used for centuries as emetics, diuretics, and arrow poisons [9].

Bufadienolides are one type of CTS comprising two substances, bufalin (BF) and marinobufagenin. They were first primarily isolated from the skin of Bufo toads (formerly *Bufo marinus,* now *Rhinella marina* L.) and were used to treat heart diseases in traditional Chinese medicine [1]. The use of digitalis, ouabain, and strophanthin glycosides to slow the rate and strengthen the contractility of the heart is one of the most important strategies to combat cardiac failure. Among these agents, digitalis glycosides are the most widely used compounds. Their positive inotropic and negative chronotropic effects on cardiomyocytes are related to the coupling between Na^+^/K^+^ ATPase and Na^+^/Ca^2+^ exchanger through their intracellular concentration.

There are extraordinary arsenals of chemicals in the animal kingdom that are critical for their survival, defending against predators and hunting [10,11]. The mucous glands are a significant source of bioactive compounds due to their evolutionary adaptations [12,13]. The skin of toads is distinguished by serous and mucous glands which release a variety of chemicals that aid in the defense against poisoning predators [13,14]. Mucous glands are many and slightly smaller. They are crucial for maintaining the skin’s pH, moisturizing it, and secreting a cocktail of bioactive compounds [15]. In addition, cardiotonic steroids have a wide spectrum of pharmacological activities, among them cardiotonic, anti-arrhythmic, antidiabetic, immunomodulatory, antibacterial, antifungal, antiprotozoal, antiviral, antineoplastic, sleep inducing, analgesic, contraceptive, endocrine activity, behavioral changes, and wound healing [16]. Clinically, CTS have demonstrated potent inhibition of cardiac Na^+^ and K^+^-ATPase (NKA), the integral membrane protein that maintains ionic gradients in cells, and therefore, they have potential activity in heart failure treatment [17]. This interaction inhibits the enzyme, stops the transport of sodium to the outside of the cell and potassium to the inside of the cell, and activates the calcium pump, which increases cardiac contraction [18]. The serous granular glands are intimately associated with the defense mechanisms against predators. These glands are scattered across the skin’s surface or clustered next to each other, similar to parotoid glands. Although the genus *Bufo* has parotoid glands mostly on the head, neck, or shoulder [19], the parotoid glands are more often seen as paired protuberances in the postorbital place in the Anura [15,20]. Targeting the Na^+^/K^+^ ATPase pump has been a focus of numerous research groups to reveal the pharmacological impacts of CTS, including cardiotonic, anticancer, anti-inflammatory, and antiviral activities [21,22,23,24].

### 1.1. Synthesis and Factors Affecting Poison Composition

Toads employ a complex series of protective processes, beginning with expanding the lung, elevating the body, and spraying the venom. External pressure is likely the most critical factor in the release of parotoid venom. Venom jets are not released from the parotoids until adequate mechanical pressure is applied. Additionally, lung inflation contributes to the expansion of lung pressure, which may be moved over to the parotoid floor and passed to the bottle-shaped parotoid glands. Consequently, the secretion is forced out via the duct slit, enabling venom jets to spray out [20].

While food has a role in the formulation of toad poison, bufadienolides can be produced through the metabolism of bacteria that colonize the parotoid glands. Bufadienolides can undergo various structural transformations, resulting in a high degree of structural diversity in this family of metabolites [25]. Early in larval development, bufadienolides are produced [26]. Toads of common occurrence *Bufo bufo* L. produce toxic or unappealing bufadienolides for various predators [27]. Bufadienolides have been reported to be more diverse and abundant in toad larvae in natural populations with a higher competitor density [28] and correspondingly increased when tadpoles have been subjected to food limitations in the laboratory [26].

If threatened, *Rhinella jimi* Stevaux typically shows a stereotyped defensive behavior, which includes inflating the lungs and generating a stiffened and voluminous posture, usually accompanied by a head-butting pattern, wherein the animal tilts its body in the direction of the dangerous and potentially agent, exposing one of the parotoids [13]. Head-butting appears to be a major component of a toad’s defense strategy, since exposing the parotoid improves the likelihood of the animal’s venom being triggered and delivered before the predator can bite it.

### 1.2. Biosynthesis of CTS of Toad Origin (Bufadienolides)

Despite substantial progress in the structure elucidation and mechanisms of the action of CTS (bufadienolides), the biosynthesis of these endogenous CTS is still poorly understood. Since the enzymes responsible for the synthesis of specific bufadienolides have not yet been identified, the ability to develop knockout and/or overexpression models is also limited so far. Chiadao and Osuch (1969) evaluated bufadienolide biosynthesis via injection of some precursors labeled with ^14^C or tritium into toads (*Bufo marinus*), which included Cholesterol-4-^14^C, pregnenolone-4-^14^C, methyl 3β-formoxy-5β-cholanate-24-^14^C, sodium 3β-hydroxy-5β-cholanate-24-^14^C, methyl 3β-acetoxy-5-cholenate-^3^H, methyl 3β-hydroxy-5α-cholanate-^3^H, and methyl 3α-hydroxy-5β-cholanate-24-^14^C. It was found that both 5α and 5β, 3β-hydroxycholanate derivates were more competent precursors for biosynthesis, as shown in Figure 2 [29]. However, according to Siperstein et al., cholesterol is the precursor responsible for the synthesis of marinobufagin by the *B. marinus* toad [30] (Figure 3). Moreover, [2-^14^C] mevalonic acid, [20-^14^C] pregnenolone, and [20-^14^C] cholesterol were evaluated as precursors for bufadienolides biosynthesis in *Bufo pnracnemis* toads by Proto et al. It has also been reported that both pregnenolone and mevalonic acid were poor precursors in winter (natural hibernation period), but mevalonic acid was involved in the summer (active life of the toad), while cholesterol was potent in both seasons of the year. The outcomes also indicated that the double unsaturated δ-lactone ring of the bufadienolides was directly derived from the cholesterol side chain without the compound’s prior conversion into pregnenolone [31].

### 1.3. Structure and Morphology of Glands Secreting Venom in Toad

Poison glands have been generated in anuran amphibians by a singular multinucleated cytoplasm mass that produced a secretory syncytium. Lumen did not exist, and a matrix of syncytial cytoplasm surrounded the gland. After being generated in the syncytial periphery, the granular secretions were maintained across the cytoplasm [32]. Individuals or clusters of glands can be seen in certain parts of the body. They resemble warts, which are frequently observed on the dorsal epidermis of toads (*bufonids*) if gathered [13]. If they occur in massive quantities, they can produce prominent protrusions called macro-glands [13,33], amongst which even the parotoids are unquestionably the most prevalent. Due to the huge size of the parotoid individual glands, these cutaneous systems can produce and store considerable quantities of poison for chemical defense against predators [20,34]. Parotoids in the *Rhinella* genus toads are histologically described as huge stockpiles of gigantic poison glands implanted into the dermis and grouped side by side to form honeycomb-like structures [35,36,37]. Salamanders [38], phyllomedusin-producing tree frogs [36], frogs [34], and toads [20] all have parotoids. They have a duct that is surrounded by a layer of myoepithelial cells. As compared with skin poison glands, the epithelial that coats the duct internally is extremely dense, obstructing the ductal canal and occasionally leaving only a tiny split in the middle [20,36,38,39]. Distinguishable mucous glands are also referred to as accessory glands, because they encircle each other [20,36,39]. Two bufadienolides have been isolated from the secretion of *Rhinella jimi* parotoid macro-glands, and both showed action against *Leishmania chagasi* Nicolle [40,41,42].

## 2. Chemistry of Cardiotonic Steroids (CTS)

The chemistry of steroids has been investigated thoroughly, including steroid glycosides isolated from toads, sponges, star fishes, fungi, and echinoderms [43,44,45,46]. However, toads cardiotonic steroids are specifically difficult to identify, and therefore, are less explored as compared with their terrestrial analogues.

The term CTS is commonly use as a synonym for cardiac glycosides (CGs), owing to their commonly glycosylated steroid nuclei and their potential biological activity. They are a unique category of phytosteroids that can be classified into two types based on their source and chemical feature: (1) cardenolides and (2) bufadienolides (Figure 1), isolated from the skin and the parotoid gland of toads [47].

Structure activity relationships (SAR) of CGs have been studied based on natural and semisynthetic analogues, their abilities to bind with sodium pump receptors, i.e., Na^+^/K^+^ ATPase pump, and subsequent cardiotonic effects. The SARs were summarized by Melero et al. and Gopalakrishnakone et al. [21,48]. The understanding of SAR has helped the synthesis of semisynthetic cardenolides with more potency than the native molecules [49].

Cardenolides (C_23_) and bufadienolides (C_24_) both bear common structural features such as the *cis*-fused C/D rings with a hydroxy group positioned at 14*β*. The major difference between them involves the lactone ring at the 17*β* position. Hence, the basic sterane (cyclopentanoperhydrophenanthrene) structure in this unique *cis*-*trans*-*cis*-configuration makes the CTS distinct from other steroidal compounds such as bile acids, sterol, or steroid hormones [50].

Cardenolides possess an unsaturated lactone ring substituent, i.e., *α*, *β*-unsaturated five-membered butyrolactone ring (but-2-en-4-olide ring), while bufadienolides have a six-membered unsaturated α-pyrone ring (penta-2,4-dien-5-olide ring) [47,51], since saturation of the lactone ring attenuates the CT bioactivity [50]. In addition, some CTS are usually found as sugars, where the aglycone part (cardiotonic steroid) is linked at the 3*β*-OH group to mono- or oligosaccharide sugar chains (maybe one, two, three, or four sugars) named glycone or sugar moiety. The sugars are thought to be _D_-glucose, _L_-rhamnose, _L_-arabinose, and _D_-xylose (Figure 1) [52,53]. In general, there are homo-oligomer and hetero-oligomer CTS. For example, scillaren A is a hetero-oligomer bufadienolide, which is glycone-structured rhamnose/glucose, while digoxin is a homo-oligomers, which is glycone composed of three units of digitoxose [51]. The most common glycosidic bonds connecting saccharide units are *O*-glycosidic bonds in which the oxygen from a hydroxyl group becomes linked to the carbon atom [54]. This glycone moiety enhances the water solubility of the glycoside and its ability to bind to its molecular target, i.e., Na^+^/K^+^-ATPase or the sodium pump in the heart muscle [50,55]. Hence, there are many sites for interaction via hydrophobic binding by steroidal nucleus and sugar moiety, in addition to the electrostatic and hydrogen bonding by the lactone ring [48,56].

Nevertheless, terrestrial plants and animal CTS are still underestimated and, consequently, less medicinally explored. The well-known CGs include those isolated from frog venoms. These compounds are mainly bufadienolides.

Toads’ (34 genera and 410 species [57]) CTS are of the bufadienolide type and occur either as free genins (bufagins) or glycosides bound to suberylarginine (bufotoxins) [48].

These toxins are found in the skin of parotoid glands and, consequently, can be extracted from dried secretion of the auricular and skin glands of toads. With the aid of metabolomics, specifically with metabolite profiling with ultra-high-performance liquid chromatography-quadrupole time-of-flight mass spectrometry (UHPLC-Q-TOF-MS) of toad secretions, 31 bufadienolide metabolites have been identified [58]. Some examples of the CTS are shown in Figure 4. The major differences among these compounds are the hydroxylation at 11*α* either alone as in gamabufagin isolated from *Bufo japonicus* Temminck & Schlegel and 11α-hydroxytelocinobufagin in *Rhinella marina* L. or at 19*β,* as in 11α, 19-dihydroxy-telocinobufagin in *R. marina*. Other structural modifications may also involve *14β, 15β*-epoxy ring formations as in marinobufagenin and resibufogenin found in *Bufo rubescens* (now *Rhinella rubescens* Lutz) and *Bufo gargarizans* Cantor, respectively [58].

Based on Kamano et al. and Gopalakrishnakone et al., structural modifications in the BF molecule can increase or decrease the cytotoxic activity against primary liver carcinoma cells PLC/PRF/5. For example, the acetylation of the 3*β*-hydroxyl group, 11α-hydroxyl substituent, and aldehyde group at 19 have been shown to increase the antitumor effect, while 5*β*-hydroxyl, 16*β*-acetoxy, and 19-CH_2_-OH substituents decreased the activity [21,59]. In addition, microbial biotransformation of cinobufagin has been reported by the filamentous fungus *Alternaria alternata* (Fr.) Keissl. (Fr.), explained by the production of 12*β*-hydroxyl and 3-oxo-12*α*-hydroxyl derivatives. The products exhibited decreased but still significant in vitro cytotoxic activities [60].

## 3. Antiviral Activity

In particular, the antiviral effects have recently attracted interest as compared with the well-documented antitumor activity [61,62,63,64]. Inhibition of the transmembrane Na^+^/K^+^ ATPase protein affects cell host intracellular signaling processes and viral life at different levels, resulting in inhibition of the viral genome expression, replication, protein translation, release, and entry. Nevertheless, these activities have been thoroughly investigated for CTS [65,66,67,68].

The CTS with antiviral activity are reported in Table 1. BF is active against coronaviruses, i.e., severe acute respiratory syndrome coronavirus 2 (SARS-CoV-2) and Middle East respiratory syndrome coronavirus (MERS-CoV) via inhibition of Src signaling mediated by ATP1A1 and virus entry into the host cells [66,69]. In a comparative study among CTS, bufalin, cinobufagin, and telocinobufagin had high anti-MERS-CoV activity, and bufalin had the most potent anti-SARS-CoV and SARS-CoV-2 activity [70]. The most potent anti-coronaviral compounds were, cinobufagin, telocinobufagin, followed by bufalin. They acted by downregulating the cell death-related genes as well as the immune- and inflammatory-related genes to balance the levels of C/D-class small nucleolar RNAs (SNORDs) and H/ACA small nucleolar RNAs (SNORAs). Additionally, the toxicities of bufalin, cinobufagin, telocinobufagin, bufotalin, and cinobufotalin were tested in vivo by Jin et al., 2021, suggesting that bufalin had the highest anti-coronaviral activity as well as the strongest toxicity. Thus, cinobufagin and telocinobufagin were selected for their high anti-coronavirus activity and low toxicity [70]. Moreover, cinobufacini (BF and cinobufagin) has been reported to inhibit mRNA expression of the hepatitis B virus (HBV) [71]. BF (15 nM) also has a higher potency than ouabain and digoxin as an anti-HSV to reduce viral yield by 90% [68]. Cinobufagin and resibufogenin showed 50% inhibitory concentrations (IC_50_) against enterovirus 71 (EV71) infection in vitro at 10.9 ± 2.4 and 218 ± 31 nM, respectively. EV71 is a pathogen affecting hand, foot, and mouth disease (HFMD) that induces CNS inflammation and life-threatening systemic complications, i.e., cardiorespiratory failure. Viral protein synthesis are likely to be targeted by both compounds [72]. BF and cinobufagin also inhibit HIV-1 with an IC_90_ at 15 and 40 nM, respectively [73].

## 4. Anticancer Activity

The anticancer activities of cardiotonic steroids (CTS) from different toad species are shown in Table 2. Cardiotonic steroids or cardiotonic glycosides are characterized by their abundance in nature, diversity of structure, potential for chemical modification, and wide use in heart failure management. CTS act by binding to the extracellular surface of Na^+^/K^+^-ATPase. The altered expression of the sodium pump subunits in different cancers strongly suggests that targeting Na^+^/K^+^-ATPase represents a novel means to fight the growing number of cancers [74]. Cardiotonic action occurs via inhibition of Na/K-ATPase, mediating cardiac muscle contraction [75]. Inhibition of Na/K-ATPase increases the influx of Na^+^ followed by a decrease of K^+^ efflux in heart cells. Increases in Ca^2+^ ions raise the cardiac contractile force, since more Ca^2+^ is available for the involved proteins. CGs trigger the accumulation of intracellular Ca^2+^ levels leading to an increase in muscle tone and the circulating blood volume per minute, leading to the control of heart rate and stroke volume [76]. Numerous studies have evaluated changes in the transmembrane transport of cations during the course of malignant cell transformation, due to an increase in Na^+^/K^+^-ATPase activity [77,78,79,80]. There is evidence that these kinetic changes in Na^+^/K^+^-ATPase activity are already present at very early stages of tumorigenesis, even before the morphological alteration and the abnormal appearance of tumors [81,82]. CTS stimulate protein tyrosine phosphorylation and a number of growth-related pathways in a cell- and tissue type-dependent manner [83,84,85,86,87,88]. CTS exhibit in vitro cytotoxic and cytostatic effects against various human cancer cell lines, which is attributable to their ability to induce cell-type-specific cell death modalities [89]. Additionally, selected CGs have entered phase I and II of clinical trials for the treatment of solid tumors with satisfactory safety and efficacy [90,91].

The antiproliferative activity of CGs on different types of cancer cell lines, including those of breast cancer, prostate cancer, pancreatic cancer, leukemia, neuroblastoma, and melanoma, have been previously described [86,92]. The predominant theory for the mode of action of CGs in tumor cells focuses on their binding to the α-subunit of Na/K-ATPase, leading to changes in pumping activity, which increases the intracellular Na^+^ levels and depletes the K^+^ levels. Consequently, the intracytoplasmic Ca^2+^ levels increase owing to exacerbation of mitochondrial Na^+^/Ca^2+^ exchange. Other theories have suggested that CGs may interact with the plasma membrane via the steroid nucleus, causing a change in membrane fluidity and indirectly affecting the function of several membrane proteins and receptors [93,94]. The sodium pump is composed of multifunctional groups of α and β subunits. The α-1 subunit of the sodium pump is overexpressed in some types of cancer, including non-small cell lung cancer (NSCLC), renal carcinoma, glioma, and melanoma, whereas the alfa-3 subunit is overexpressed in colon carcinoma [93,95,96]. In addition, in mice, α-2, α-3, and α-4 isoforms are naturally susceptible to CGs. In rodents, the α-1 isoform is resistant to the binding of ouabain and, in humans, the α -1 isoform is sensitive to CGs and may, therefore, play a key role in the signal transduction pathways [97].

CGs can induce several signal transduction pathways such as inhibition of DNA topoisomerase II [98]; inhibition of the nuclear factor-κ light-chain-enhancer of activated B cell (NF-κB)-mediated pathways; changes in cell cycle, specifically blocking S and G_2_/M phases; inhibition of interleukin (IL)-8 production [99]; and modulation of the myeloid cell leukemia-1 (Mcl-1) as an essential factor for cell death [5,100]. CGs and particularly ouabain are involved in metastatic cascades by inhibiting the migration of H292 lung tumor cells via the suppression of regulatory migration proteins, such as focal adhesion kinase (FAK) and Akt [101]. Furthermore, digitoxin has been shown to inhibit angiogenesis in human umbilical vein endothelial cells (HUVEC) and promoted FAK activation by several pro-angiogenic stimuli [102]. Gamabufotalin has been shown to inhibit vascular endothelial growth factor (VEGF)-triggered HUVEC proliferation, migration, and invasion in vitro by suppressing the VEGF receptor-2 signaling pathway [103].

BF-induced apoptosis in human leukemia cells [104] is mediated by ERK-kinase cascade which is excessively activated in order for BF-mediated apoptosis to occur [105,106,107]. Treatment of human leukemia THP-1 cells with BF-induced inflammatory cytokine interleukin-1 beta (IL-1β) and tumor necrosis factor-α (TNF-α). After treating the cells with an inhibitor of ERK, i.e., PD-98059, the cytokine production was attenuated, suggesting that the ERK pathway was responsible for the inflammatory response induced by BF [105].

BF exerts an antitumor effect by inducing apoptosis and triggering autophagic cell death in various human cancer cells. The anti-inflammatory activities of BF are also important for its antitumor function [107]. BF has been shown to inhibit proliferation and induced mitochondria-dependent apoptosis in U2OS and Saos-2 cells [108]. Furthermore, a mechanistic investigation demonstrated that BF was able to significantly decrease Mcl-1 expression level and modestly decrease Bcl-XL expression level. The downregulation of these anti-apoptotic proteins has been well correlated with deactivation of transcription factor STAT3. BF and cinobufagin (1–10 μM) inhibited the proliferation and induced cell apoptosis of androgen-dependent (LNCaP) and independent (DU145 and PC3) prostate cancer cell lines [109]. Qi et al. [110] found that BF and cinobufagin induced apoptosis and increased the proportion of apoptotic cells. This apoptotic induction was associated with an increase in Fas, Bax, and Bid expression; a decrease in Bcl-2 expression; disruption of the mitochondrial membrane potential; release of cytochrome c; activation of caspase-3, -8, -9, and -10; and the cleavage of poly (ADP-ribose) polymerase (PARP), which indicated that BF and cinobufagin induced apoptosis through both Fas and mitochondria-mediated pathways [111]. BF also inhibited APL cell proliferation in a time- and dose-dependent manner and induced NB4 cell apoptosis accompanied by downregulation of survivin and activation of caspase-3. The MEK/ERK signaling pathway was negatively regulated in BF-induced apoptosis in the NB4 human leukemia cell line. BF enhanced ATRA-induced differentiation in NB4 cell line and primary culture in APL cells [112], and also induced MGc-803 cell death characterized by apoptotic phenotype DNA content changes and chromosome DNA fragmentation. BF triggered cell cycle arrest in G1 phase, leading to the induction of apoptosis of gastric cancer cells [113]. A distinct study has concluded that BF-induced apoptosis by altering the expression of apoptosis-related genes c-Myc and Bcl-2 [114], and the activation of mitogen-activated protein kinase (MAPK) may be involved in BF-induced apoptosis in U93T cells [115]. Overexpression of Bcl-2 inhibited BF-induced MAPK activation and the subsequent AP-I activation and cell apoptosis in U937 cells [113].

BF was able to suppress the migration and invasion of prostate cancer cells through HOTAIR, the sponge of miR-520b [115]. In addition, the same natural product inhibited human breast cancer tumorigenesis by inducing cell death through the ROS-mediated RIP1/RIP3/PARP-1 pathways [116]. BF also attenuated the proliferation of breast cancer MCF-7 cells in vitro and in vivo, by inhibiting the PI3K/Akt pathway, and the proliferation of breast cancer MCF-7 cells in vitro and in vivo by inhibiting the PI3K/Akt pathway [117]. The toad skin extract cinobufatini inhibited migration of human breast carcinoma MDA-MB-231 cells in a model stromal tissue [118], while key members of bufadienolides, i.e., BF, bufotalin, and gamabufotalin, have significantly sensitized human breast cancer cells with different status of ER-α to apoptosis induction of TRAIL, as evidenced by enhanced Annexin V/FITC positive cells (apoptotic cells), cytoplasmic histone-associated DNA fragments, membrane permeability transition (MPT), caspases activation, and PARP cleavage [119]. A BF derivative has exhibited stronger apoptosis-inducing effect than BF in A549 lung cancer cells and lower acute toxicity in mice [120]. The original natural product induced apoptosis of lung cancer cells via the regulation of the PI3K/Akt pathway. [121] used the A549 human lung adenocarcinoma epithelial cell line as an experimental model to evaluate the effects of BF in lung cancer chemotherapy. BF suppressed colorectal cancer cell growth through promoting autophagy in vivo and in vitro [122]. In addition, it was able to inhibit hTERT expression and colorectal cancer cell growth by targeting cpsf4 [123], to reverse ABCB1-mediated drug resistance in colorectal cancer [124], and to regulate mTOR and ERK signaling pathways in gastric cancer cells [125]. BF treatment could decrease miR-298 expression. Previously, we have also shown that the deletion of miR-298 contributed to BF-induced apoptosis in gastric cancer cells by targeting BAX, an apoptosis protein [126]. BF revealed anticancer effects on human hepatocellular carcinoma HepG2 cells [127].

The bufadienolides are a group of steroid compounds belonging to CGs, a class of circulating substances [128,129,130,131]. Marinobufagenin (MBG), an endogenous mammalian cardiotonic agent [132,133], is currently the most commonly used in drug research. MBG has been implicated in various physiological conditions [134] and appeared to be associated with pathophysiological events in animals [135,136,137] and humans [138,139,140]. MBG inhibits glioma growth in vivo and in vitro through the sodium pump α1 subunit and ERK signaling-mediated mitochondrial apoptotic pathways, and activates ERK/NF-κB signaling, activating the caspase signaling cascade through accelerating the release of cytochrome C, inducing apoptosis. These data support the potential use of MBG as an anticancer agent [141].

Resibufogenin (RB) is a commonly used natural medicinal compound extracted from Asiatic toad *Bufo gargarizans* Cantor. This compound is known for its analgesic, anti-inflammatory, anticancer, and anti-radiation properties, as well as its cardio protective and anesthetic activities. RB effectively inhibits cell viability, and induces apoptosis, caspase-3, and caspase-8 activities in MGC-803 cells by suppressing the PI3K/AKT/GSK3β signaling pathway. Therefore, it may be considered as a possible treatment for gastric carcinoma [142]. RB suppresses ovarian cancer growth and glycolysis in vitro and in vivo. Moreover, the downregulation of PIM1 by RB plays a key role in the anticancer activities of RB [143]. RB suppresses colorectal cancer growth and metastasis, activating RIP3 and phosphorylating MLKL, leading to necroptosis [144]. RB suppresses transforming growth factor-β-activated kinase 1-mediated nuclear factor-κB activity through protein kinase C-dependent inhibition of glycogen synthase kinase 3 [145].

## 5. Anti-Inflammatory Activity

Some research has suggested that cardiotonic steroids, including bufalin, have immunomodulatory properties [153], as they interact with numerous inflammatory responses such as vascular permeability, cell migration, and proinflammatory cytokines [153,154]. The main CTS with anti-inflammatory activity are shown in Table 3. Carvalho et al. [155] demonstrated that the anti-inflammatory activity of obufagenin in vivo and in vitro exposed a novel endogenous function for this steroid in mammals, owing to its ability to reduce cytokine levels in peritoneal macrophage culture (Table 2).

Bufalin has a long history of being used as an anti-inflammatory medicinal drug in China and other Asian countries [156]. Wen et al. [157] investigated bufalin’s analgesic and anti-inflammatory properties in vivo, suggesting that it could be a possible drug treatment for inflammatory diseases. They observed that bufalin effectively suppressed NF-κB activation in vivo by preserving IκBα levels, decreasing the nuclear translocation of NF-κB p65, and inhibiting downstream proinflammatory mediators. Ye et al. [158] demonstrated that bufalin suppressed the nuclear translocation of NF-κB in response to TNF in vitro. In addition, bufalin has been shown to control NF-κB activity [159], a key regulator of the inflammatory process that plays a significant role in inflammation. It modulates the expression of proinflammatory mediators, including cyclooxygenase-2, IL-6, inducible nitric oxide synthase, and TNF, IL-1*β* [160]. NF-κB signaling is the ideal therapeutic target for inflammation pathogenesis. Furthermore, the bufalin substitutes, ouabain and digoxin, have potent anti-inflammatory properties [161,162]. Ye et al. [158] revealed that bufalin suppressed tumor necrosis factor (TNF) signaling in human 293T cells via interfering with NF-κB nuclear translocation. Zhakeer et al. [9] demonstrated that bufalin attenuated hyperresponsiveness and suppressed increases of total inflammatory cells in a mouse asthma model. The levels of IL-4, IL-5, and IL-13 in serum were significantly reduced. Cell infiltration, goblet cell hyperplasia, IκBα degradation from NF-κB, and the level of phosphorylated p65 protein levels in the lung tissues were all decreased, indicating that it may mediate its anti-inflammatory effects via inhibiting NF-κB activity. Bufalin may inhibit the activation of NF-κB and may reduce the production of its downstream proinflammatory mediators during acute inflammation [157].

## 6. CTS with Antiviral, Anticancer, and Anti-Inflammatory Properties

The potential pharmaceutical properties of bufadienolides isolated from toad venom have been investigated recently. Anticancer activity of bufalin has been seen against breast, liver, and gastric and leukemia cancer cells, as listed in Table 1. It has been indicated that bufalin inhibited tumor growth through apoptosis induction by multiple pathways [165]. In in vivo studies, bufalin suppressed human hepatocellular carcinoma (HHC) cell growth and induced apoptosis by activating Bax without noticeable toxicity [114]. In another study, nude mice injected with HCCLM3-R cells were studied after bufalin treatment. Significant antitumor activities, manifested with the regress noticed in the metastatic growth in parallel with the inhibition of AKT/GSK3/catenin/E-cadherin signaling pathways, were evident [166]. Bufalin inhibited tumor growth by inducing cell apoptosis through the intrinsic apoptotic pathway. A reduction in tumor size of the human lung cancer cell line, NCI-H460, was confirmed after bufalin treatment without significant drug-related toxicity [167].

Anti-inflammatory and anticancer effects of bufalin have been illustrated through the inhibiting NF-B pathway, which is an important pathway in both anti-inflammation and cancer [168,169]. Bufalin reduces hyperresponsiveness, and inhibits the OVA-induced activation of inflammatory cells, including macrophages, eosinophils, lymphocytes, and neutrophils, as well as cytokines, including IL-4, IL-5, and IL-13. Additionally, a reduction in inflammatory cell infiltration and goblet cell hyperplasia and blockage of NF-B have been noticed [9]. The anti-inflammatory and analgesic effects of bufalin have been studied in a carrageenan-induced paw oedema model. Bufalin downregulated the expression of nitric oxide synthase (iNOS), cyclooxygenase-2 (COX-2), interleukin-1 (IL-1), interleukin-6 (IL-6), and tumor necrosis factor (TNF), to which the inhibitory effect on the master switch of NF-B signaling was attributed [157]. Moreover, bufalin had high anti-MERS-CoV activity, and had the most potent anti-SARS-CoV and SARS-CoV-2 activity [70].

## 7. Conclusions

Using natural products as therapeutic agents has gained much attention, therefore, it is noteworthy to document the recent updates in the literature with particular emphasis on the possible applications of cardiotonic steroids in the promotion of anticancer and antiviral strategies. Limited numbers of CTS have been expected to have anticancer activity. Cardenolides, mainly attributed to digitalis and digoxin, have potential effects in the treatment of congestive heart failure and atrial fibrillation by binding to the extracellular membrane and activating Na^+^/K^+^-ATPase. Targeting Na^+^/K^+^-ATPase and altering the expression of the sodium pump subunits represents a novel means to fight the growing number of cancers. For instance, marinobufagenin has been shown to inhibit the growth of glioma cancer in vivo and in vitro via the sodium pump α1 subunit and ERK signaling-mediated mitochondrial apoptotic pathways. Additionally, γ bufotalin has been reported to suppress vascular endothelial growth factor (VEGF)-triggered proliferation, migration, and invasion by suppressing the VEGF receptor-2 signaling pathway in vitro. Similarly, resibufogenin (RB) is a known anticancer agent that acts via the inhibition of cell viability, induces apoptosis, caspase 3, and caspase 8 activities in MGC 803 cells by suppressing the PI3K/AKT/GSK3β signaling pathway. Finally, bufalin (BF) is active against corona viruses, i.e., SARS-CoV-2 and MERS-CoV, via inhibition of Src signaling mediated by ATP1A1, and thus, blocking entry of the virus into the host cells. As well, BF and cinobufagin inhibit mRNA expression of the hepatitis B virus (HBV) and HIV-1. Furthermore, cinobufagin and resibufogenin have been shown to prevent 71 (EV71) infections from enterovirous. In the current review, we illustrate the significance of cardiotonic steroids as potential anticancer, anti-inflammatory, and antiviral candidates.

Further studies are suggested to screen the efficacy of CTS isolated from toad species from different geographical regions, and under different environmental conditions to ensure stability of their chemical composition. Additional molecular mechanisms and modeling, drug safety, and quality control are needed to motivate the future implications of CTS as novel therapeutic agents that will certainly contribute to the development of pharmaceutical industry.

## Figures and Tables

**Figure 1 molecules-27-06586-f001:**
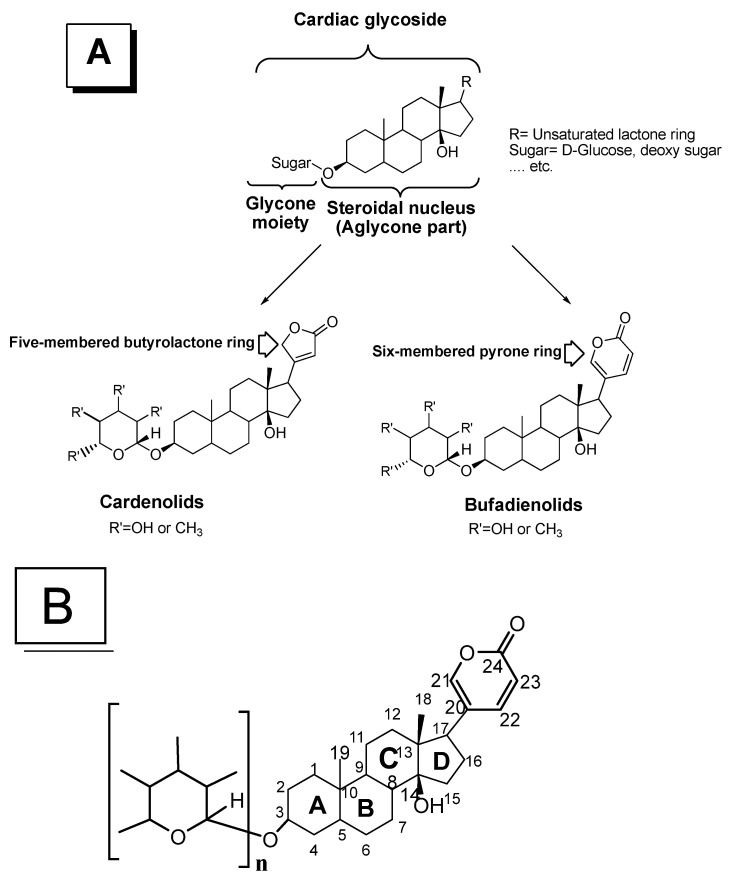
(**A**) The basic steroid aglycone unit of cardiotonic steroids and their types, cardenolides and bufadienolides; (**B**) numbered chemical structure of the bufadienolide aglycone or steroidal nucleus, i.e., the main type of cardiotonic steroids (n denotes the number of the sugar monomer in the cardiac glycoside structures (*n* = 1–3)).

**Figure 2 molecules-27-06586-f002:**
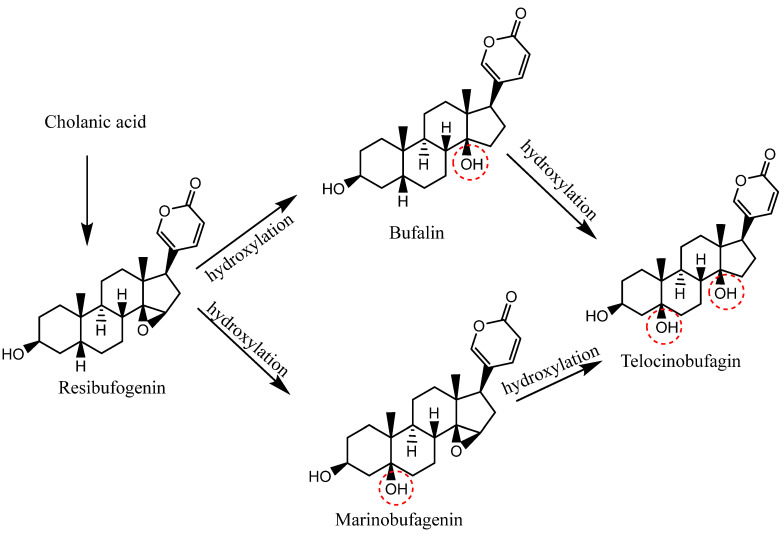
Proposed mechanism of the biosynthesis of bufadienolides by toads (*Bufo marinus*) [29].

**Figure 3 molecules-27-06586-f003:**
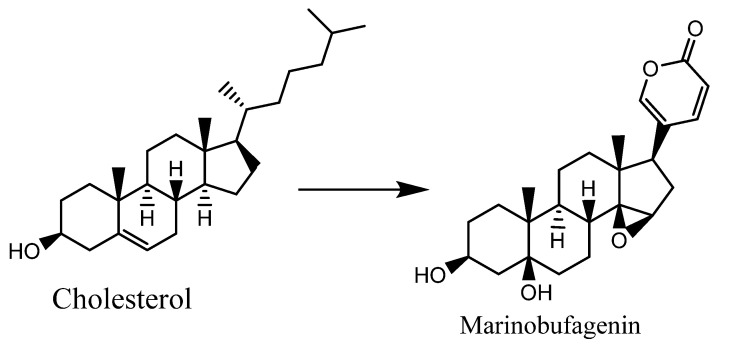
Proposed mechanism of the biosynthesis of marinobufagenin by toads (*Bufo marinus*) [30].

**Figure 4 molecules-27-06586-f004:**
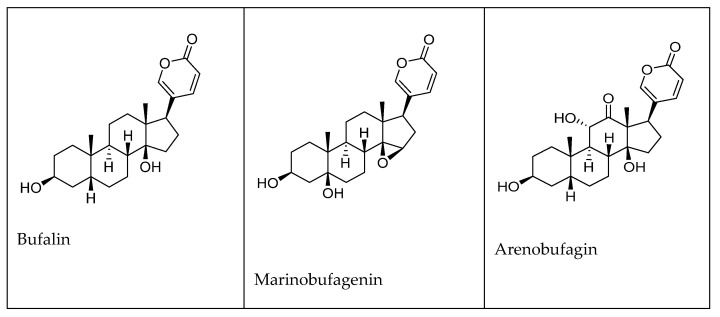
Chemical structures of most common compounds isolated from venom of toads.

**Table 1 molecules-27-06586-t001:** Antiviral activities of cardiotonic steroids (CTS) from different toad species.

Toad Species	CTS	Target	Ref.
*Bufo gargarizans* Cantor	Bufalin	Anti-hepatitis B above 10^-2^ µM Anti-HIV, IC_50_ = 5 nm Anti-MERS-CoV in vero cells (IC_50_ = 0.018 µM) after 24 h Anti-MERS-CoV Calu-3 human lung cells (IC_50_ = 0.544 µM) (in vitro)	[70,71,73]
*Bufo gargarizans*	Cinobufagin	Anti-hepatitis B above 10^-1^ µM Anti- enterovirus 71 (EV-71) IC_50_ = 10.94 nmol/L Anti-HIV, IC_50_ = 24 nm Anti-MERS-CoV in vero cells IC_50_ = 0.017 µM after 24 h Anti-MERS-CoV Calu-3 human lung cells IC_50_ = 0.616 µM (in vitro)	[70,71,72,73]
*Bufo gargarizans*	Resibufogenin	Anti-enterovirus 71 (EV71) IC_50_ = 218 nmol/L Anti-MERS-CoV in vero cells (IC_50_ = 1.612 µM) after 24 h Anti-MERS-CoV Calu-3 human lung cells IC_50_ = 15.970 µM (in vitro)	[70,72]
*Bufo gargarizans*	Telocinobufagin	Anti-MERS-CoV in vero cells IC_50_ = 0.027 µM after 24 h Anti-MERS-CoV Calu-3 human lung cells IC_50_ = 0.465 µM (in vitro)	[70]
*Bufo gargarizans* Cantor	Bufotalin	Anti-MERS-CoV in vero cells IC_50_ = 0.063 µM Anti-MERS-CoV Calu-3 human lung cells IC_50_ = 1.630 µM (in vitro)	[70]
*Bufo gargarizans* Cantor	Cinobufotalin	Anti-MERS-CoV in vero cells IC_50_ = 0.23 µM after 24 h Anti-MERS-CoV Calu-3 human lung cells IC_50_ = 3.958 µM (in vitro)	[70]

**Table 2 molecules-27-06586-t002:** Anticancer activities of cardiotonic steroids (CTS) from different toad species.

Toad Species	Detected CTS	Target Cell and Mechanism of Action	Ref.
*Bufo gargarizans* Cantor/ *Bufo melanostictus* Suhneider (Toad venom)	Arenobufagin	HepG2 cells (IC_50_ = 20.24 ± 3.84 nM) HepG2/ADM cells (IC_50_ = 7.46 ± 2.89 nM) HL-60 cells (IC_50_ = 27.70 ± 8.77 nM) after 72 h Induces apoptosis and autophagy, inhibition of the PI3K/Akt/mTOR pathway (in vitro, in vivo)	[146]
*Bufo gargarizans*	Arenobufagin	Non-small cell lung cancer cell (A549) IC_50_ = 12.530 ng/mL after 72 h Induces apoptosis in A549 cells with the enhanced expression of cleaved PARP (poly ADP-ribose polymerase (in vitro)	[147]
*Bufo melanostictus Schneider*	Arenobufagin	Lung cancer (A549) At (0.5, 1 and 2 nM) inhibited the mobility of A549 cells (59.9%, 41.1%, and 24.7%, respectively) At (0.5, 1, and 2 nM) inhibited the mobility of H1299 cells (72.3%, 47.4%, and 22.4%, respectively) after 48 h Target IKKβ to inactive NFκB signaling cascade and change protein expression related to EMT (in vitro and in vivo)	[148]
*Bufo gargarizans* Cantor	Bufalin	Non-small cell lung cancer NSCLC A549 cells At 2.5–10 µM, bufalin- induced apoptosis and cell cycle arrest in G1 phase	[105,149]
		HepG2 cell (IC_50_ = 0.61 ± 0.06 µM) R-HepG2 cells (IC_50_ = 0.24 ± 0.02 µM) Induces cell cycle arrest at G_2_/M phase after 48 h (in vitro)	
*Bufo gargarizans* Cantor and *Bufo melanostictus* Schneider	Bufotalin	HepG2 cell (IC_50_ = 0.43 ± 0.07 µM) R-HepG2 cells(IC_50_ = 0.13 ± 0.01 µM) Induces cell cycle arrest at G_2_/M phase after 48 h (in vitro)	[149]
*Bufo gargarizans* Cantor and *Bufo melanostictus* Schneider	Hellebrigenin	HepG2 cells (IC_50_ = 0.40 ± 0.05 µmol/L) After 24 h (IC_50_ = 0.13 ± 0.01µmol/L) After 48 h (IC_50_ = 0.10 ± 0.01 µmol/L) After 72 h Induces cell cycle arrest at G_2_/M phase (in vitro)	[150,151]
*Bufo gargarizans Cantor*	Gamabufotalin (CS-6)	NSCLC (IC_50_ = 50 nM) Inhibit NSCLC cells growth and enhance apoptosis induction (in vitro)	[152]
*Bufo gargarizans*	Cinobufatolin	H157 cancer cells IC_50_ = 131.12 ng/mL A549 cancer cells IC_50_ = 23.08 ng/mL after 72 h (in vitro)	[147,149]
		HepG2 cell (IC_50_ = 1.58 ± 0.21 µM) R-HepG2 cells(IC_50_ = 0.74 ± 0.07 µM) Induces cell cycle arrest at G_2_/M phase after 48 h (in vitro)	
*Bufo gargarizans*	Telocinobufagin	H157 cancer cells IC_50_ = 23.60 ng/mL A549 cancer cells IC_50_ = 27.882 ng/mL after 72 h (in vitro)	[147,149]
		HepG2 cell (IC_50_ = 1.28 ± 0.19 µM) R-HepG2 cells(IC_50_ = 0.49 ± 0.05 µM) after 48 h Induces cell cycle arrest at G_2_/M phase (in vitro)	
*Bufo gargarizans*	Resibufogenin	Gastric carcinoma cells (MGC-803) (4 and 8 µM) for 24 h and 48 h Increased Bax/Bcl-2 expression, and suppressed cyclin D1, cyclin E, PI3K, phosphorylated AKT, phosphorylated GSK3β, and β-catenin protein expression in MGC-803 cells. (in vitro)	[143]

**Table 3 molecules-27-06586-t003:** Anti-inflammatory activities of cardiotonic steroids (CTS) from different toad species.

Toad Species	Detected CTS	Activity	Ref.
*Rhinella schneideri*	Marinobufagenin	Anti-inflammatory (10, 100, 1000, and 10,000 nM), decreased IL-1β, IL-6, and TNF-α levels (in vitro, in vivo)	[155]
*Bufo gargarizans* Cantor	Bufotalin	Anti-inflammatory against chronic inflammatory autoimmune disease 100 μg/kg in vivo and 200 nM in vitro inhibiting proinflammatory Th17 population and secretion of inflammatory cytokines	[163]
*Bufo gargarizans Cantor*	Bufalin	Anti-inflammatory against carrageenan-induced paw edema model (0.3 and 0.6 mg/kg, i.p.) Downregulation of expression levels of nitric oxide synthase (iNOS), cyclooxygenase-2 (COX-2), 1β (IL-1β), (IL-6), (TNF-α), and inhibited the activation of NF-κB signaling (in vivo)	[157]
*Bufo gargarizans*	Gammabufotalin	Anti-inflammatory (1, 4, and 12 (50 μM); 2, 13, and 14 (10 μM); 3 and 6 (5 μM); 5 and 8 (1 μM); 7, 9, and 11 (0.5 μM); 10 (4 μM)) Inhibits LPS-induced inflammation by suppressing myeloid differentiation primary response 88/nuclear factor-kappa B and STAT3 signal pathways. (in vivo)	[164]

## Data Availability

No applicable.

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
