# Peer review of "Chemistry and the Potential Antiviral, Anticancer, and Anti-Inflammatory Activities of Cardiotonic Steroids Derived from Toadsâ€"

_molecules, 2022, doi:10.3390/molecules27196586_

Round 1
Reviewer 1 Report
In general, the review article is interesting and allows to gather the maximum information on the question. However, it would be nice to: - specify the source and classification of cardiotonic steroids - have a figure containing the main cardiotonic steroids - have a table showing the anti-inflammatory, antiviral and anticancer activity by cardiotonic steroid;Some detailed commentaries are given below :
Page 1
Title :
The content of the manuscript focuses more on cardiotonic steroids of marine origin. Other origins are forgotten. To this end, the title should fit the content
Abstract
- Improve the background by giving a definition and/or the main chemical groups which includes the CTS. Then we can understand the link between « cardenolide » and CTS
- What is the problem? why do this study ;
- Briefly describe the methodology
- Briefly present the most convincing results
Keywords : Add the name of one or two CTS substances that have shown particular anti-inflammatory, antiviral or anti-cancer activity
Page 2, Introduction
- Paragraph 1, line 2 : Please, add reference to this statement « Steroids represent 3.3% of the marine materials content »
- Paragraph 3, lines 22 to 34 : Please, stay on your subject or go directly to the link between this part and the CTS
- Paragraph 4, lines 35 to 47 : Please, synthesize this part by focusing on what concerns the CTS
Page 3
- Before the chapitre « Structure and morphology of glands secreting venom in toad », Please, add a chapter on the origin / source of CTS in which you will describe all those that can influence the (bio)synthesis of CTS
- Please improve this section « Chemistry of cardiotonic steroids (CTS) » by showing the chemical structures of the main known CTS : digoxin, digitoxin, oleandrin, bufalin, marinobufagenin…
Page 9
- Chapter « Anti-viral activity » : Please move this sentence to the "introduction" section : « Targeting the Na+/K+ ATPase pump is a focus of numerous research groups to reveal the pharmacological impact of CTS, including cardiotonic, anti-cancer, anti-inflammatory, and anti-viral activities [45,68–70]. »
Page 9 to 12
- Chapter « Anti-cancer activity » : Please, specify IC50s and incubation times, if possible !
Page 12 to 13
- Chapter « Anti-inflammatory activity » : Please, add effective doses if possible
Page 13 : Before concluding, please add a chapter which will deal with substances having the 3 pharmacological properties together and the potential interest of such substances in therapy
Page 13: Conclusion : This section is not mandatory but can be added to the manuscript without repeating the results. It must answer the research question posed at the beginning: what is the scientific contribution of this work? what is (are) the interesting / major result(s) that should be remembered? Are there substances that have particular activities and that could be serious candidates for the development of antiviral, anti-inflammatory or anti-cancer drugs? under what conditions should the results of this work be used in therapy? What remains to be done?
Page 13, Back matter
Add the section of AknowlegdmentsAuthor Response
File attach

Reviewer 2 Report
Abstract. This is not well written, because this represents the whole of the data
Tabulated each therapeutic activity with their results
Author Response
File attach

Reviewer 3 Report
The submitted manuscript Chemistry, and the Potential Anti-viral, Anti-cancer, and Anti-inflammatory Activities of Cardiotonic Steroids meets aims and scope of „Molecules” Journal. However, in my opinion, the publication does not constitute a coherent whole and requires corrections. The information presented in the individual paragraphs of the article does not complement each other, therefore the publication can be approved only after major revision.
1. When discussing the pharmacological properties of cardiac glicosides (section 3-5), the authors mention digoxin, lanatoside C, proscillaridin A, digoxigenin, digitoxigenin, convallatoxin, bufotalin gamabufotalin, so the chemical formulas of these compounds should be included in section 2 of the publication.
2. The second section of the publication details the bufadienolides secreted by toads. If the authors wanted to focus in particular on these chemical compounds, they should include it in the title of the work and in the pharmacological part (section 3-5). Unless the authors intended it, the second section should be supplemented with a detailed description of compounds from medicinal plants, such as Digitalis purpurea, Digitalis lanata, Adonis vernalis, Convallaria majalis, Strophanthus kombé, Strophanthus gratus, Nerium oleander and Urginea maritima. In the Table 1 containing the chemical formulas of the described compounds, the names of these compounds should be at the level of the drawn chemical structures, so that there is no doubt as to what the presented formulas represent. If in the publication the authors focus on showing the pharmacological activity of the given compounds (which is suggested by the title of the article), then they should present the formulas of those that will be mentioned later.
3. I do not understand in what context the terms marine steroids (section 2 and conclusion) appear in the publication. The compounds which are described in detail here are derived from freshwater toads, or tree frogs not marine animals, and also from medicinal plants found on land. If the authors wanted to focus on cardiac glycosides occurring in marine organisms, they did not achieve such an effect in the publication with the presented content.
4. The sentence: „It is well known that cardenolides, mainly digitalis and digoxin, showed therapeutic activity against congestive heart failure.” should be corrected, because digitalis is a plant, maybe it was about raw medicinal plant material, i.e. digitalis leaf?
5. The sentence: „Vegetable or animal preparations containing CTS have been used as emetics, diuretics, and arrow poisons for centuries.” should be corrected, because these are rather herbs (medicinal plants) not vegetables.
6. In the sentence: For instance, marine-derived bacteria and fungi have provided a variety of potential pharmacological metabolites namely, polyketides, alkaloids, peptides, proteins, lipids, shikimates, glycosides, isoprenoids, and hybrids.” The term „hybrids” should be rather „chemical hybrids”.
7. Authors should answer the question in this sentence or remove the note in parentheses : „For example, the acetylation of the 3β-hydroxyl group, 11α-hydroxyl substituent, and aldehyde group at 19 (which compound is meant here?`) have increased the antitumor effect, while 5β-hydroxyl, 16β-acetoxy, and 19-CH2-OH substituents decreased the activity.”
8. The authors do not explain what they mean by the term cardiac sugars in the sentence: „In addition, regardless of the CTs type, most CTs are usually found as sugar ether, where the aglycone part is linked at the 3β-OH group to mono- or oligosaccharide-sugar chains (between one and four sugars) named glycone or sugar moiety, consisting of deoxy or cardiac sugars including 6-deoxy and 2,6-dideoxy hexoses and other common sugars (D-glucose, L-rhamnose, L-arabinose, and D-xylose).“
9. The manuscript requires the following editorial corrections:
a. in the names „α subunit” and „β subunit” there should be spaces,
b. number „2” in the names of the phase of the phase cycle – G2/M, and carbon dioxide – CO2 should be written in subscript (line 27, 114, 221, 222, 225),
c. correct the typing mistake in the word cyclopentanoperhydrophenanthrene (sterane),
d. add the list of abbreviations to the publication.
Author Response
File attach

Round 2
Reviewer 3 Report
I would like to thank the authors for modifying the publication and introducing corrections to its content.
I accept the introduced changes.
Author Response
Thanks so much for the kind evaluation